# Limited Hyperoxia-Induced Proliferative Retinopathy (LHIPR) as a Model of Retinal Fibrosis, Angiogenesis, and Inflammation

**DOI:** 10.3390/cells12202468

**Published:** 2023-10-17

**Authors:** Katia Corano Scheri, Yi-Wen Hsieh, Eunji Jeong, Amani A. Fawzi

**Affiliations:** Department of Ophthalmology, Feinberg School of Medicine, Northwestern University, Chicago, IL 60611, USA; katia.scheri@northwestern.edu (K.C.S.); yi-wen.hsieh@northwestern.edu (Y.-W.H.); eunjijeong2024@u.northwestern.edu (E.J.)

**Keywords:** retinopathy, fibrosis, angiogenesis, inflammation

## Abstract

The progression to fibrosis and traction in retinopathy of prematurity (ROP) and other ischemic retinopathies remains an important clinical and surgical challenge, necessitating a comprehensive understanding of its pathogenesis. Fibrosis is an unbalanced deposition of extracellular matrix components responsible for scar tissue formation with consequent tissue and organ impairment. Together with retinal traction, it is among the main causes of retinal detachment and vision loss. We capitalize on the Limited Hyperoxia Induced Retinopathy (LHIPR) model, as it reflects the more advanced pathological phenotypes seen in ROP and other ischemic retinopathies. To model LHIPR, we exposed wild-type C57Bl/6J mouse pups to 65% oxygen from P0 to P7. Then, the pups were returned to room air to recover until later endpoints. We performed histological and molecular analysis to evaluate fibrosis progression, angiogenesis, and inflammation at several time points, from 1.5 months to 9 months. In addition, we performed in vivo retinal imaging by optical coherence tomography (OCT) or OCT Angiography (OCTA) to follow the fibrovascular progression in vivo. Although the retinal morphology was relatively preserved, we found a progressive increase in preretinal fibrogenesis over time, up to 9 months of age. We also detected blood vessels in the preretinal space as well as an active inflammatory process, altogether mimicking advanced preretinal fibrovascular disease in humans.

## 1. Introduction

Retinopathy of prematurity (ROP) is a vaso-proliferative disorder of the eye affecting premature neonates caused by abnormal development of retinal blood vessels, and it is among the most common causes of childhood blindness. In humans, retinal vascularization is completed intrauterine, a few weeks before term birth, placing the immature, incompletely vascularized retina of preterm neonates at risk, especially with exposure to hyperoxia therapy (incubators), which disrupts neurovascular growth, leading to ROP [1]. ROP is a biphasic disease where initial hyperoxia causes an arrest of retinal vascularization, vaso-obliteration, and vessel loss. Subsequently, the metabolically active yet poorly vascularized retinal neurons experience hypoxia, and this microenvironment induces the release of factors leading to abnormal retinal neovascularization. Uncontrolled (extraretinal) proliferation of blood vessels can result in fibrovascular traction, retinal detachment, hemorrhage, and subsequent visual impairment [2]. Similar fibrovascular lesions can be observed in other ischemic retinopathies, such as proliferative diabetic retinopathy (PDR), which can also lead to significant vision loss and require surgical interventions [3]. 

The management of ROP and other ischemic retinopathies involves various treatment modalities, including laser photocoagulation, cryotherapy, and, more recently, anti-vascular endothelial growth factor (VEGF) therapy [4,5]. Despite advances in early detection and management, these retinopathies continue to pose significant challenges in the field of ophthalmology.

The most widely used rodent model to study abnormal angiogenesis and to model ROP is the classic oxygen-induced retinopathy (OIR) [6]. Unlike human ROP, this model does not develop advanced complications, with complete regression of angiogenesis and revascularization of the retina by post-natal day (P) 28. To model more severe variants of ROP, the McMenamin group introduced a chronic model of neonatal hyperoxia-induced vitreoretinopathy, the limited hyperoxia-induced proliferative retinopathy (LHIPR) [7]. This model showed quite a few similarities with the human ROP and advanced stages of diabetic retinopathy, such as preretinal fibrotic scar formation, tortuous retinal vessels, and persistent hyaloidal vessels.

We previously explored the retinal phenotype of the LHIPR model using a combination of retinal morphometry and immunopathological studies and showed that the LHIPR shows persistent retinal vascular dysfunction. We observed a delay in the retinal vascular development, vascular density, and peripheral avascular areas up to 1.5 months of age. These vascular changes were associated with retinal inflammation, pericyte loss, and preretinal fibrosis, as well as significant inner retinal thinning [8].

In this report, we wanted to further investigate the fibrovascular response in this model over a longer time span (9 months) to better understand its characteristics and the relevance of these fibrovascular lesions to those seen in complicated ROP and other ischemic retinopathies.

## 2. Materials and Methods

### 2.1. Animals

C57Bl/6J mice, both female and male (The Jackson Laboratory, Bar Harbor, ME, USA), were used for all animal experiments in this study. All the experiments were performed following the Guide for the Care and Use of Laboratory Animals of the National Institutes of Health. The protocols were approved by the Institutional Animal Care and Use Committee at Northwestern University (protocol IS00013947).

### 2.2. Limited Hyperoxia-Induced Proliferative Retinopathy (LHIPR)

The LHIPR animal model was performed as previously described [7,8]. Briefly, dams with their newly born pups were exposed to 65% oxygen from P0 to P7, using a Plexiglas chamber with an oxygen controller (ProOx 110; Biospherix, Lacona, NY, USA). Age-matched control pups were maintained at room air. To protect the mice from oxygen toxicity, dams were rotated between hyperoxia and room air every 24 h. The pups and dams were then returned to room air in conventional cages after 7 days of hyperoxia exposure, where they recovered until the termination of the experiment.

### 2.3. Histological Preparation

Enucleated globes were fixed in 4% paraformaldehyde (PFA) overnight at 4 °C and then washed in PBS for histological preparation. After fixation, eye paraffin embedding was conducted by the Mouse Histology and Phenotyping Core Laboratory of Northwestern University. Samples were then sectioned (7 μm) for further analysis.

### 2.4. Retinal Morphology Analysis

For retinal morphology analysis, samples were stained with hematoxylin and eosin (H&E) after deparaffinization and rehydration. They were imaged using a Nikon 80i Eclipse microscope (Nikon, Tokyo, Japan) equipped with a Photometrics CoolSnap CF camera (Photometrics, Tucson, AZ, USA). The images were then analyzed, and Image J software, (Image J 1.54d, National Institutes of Health, Bethesda, MD, USA) was used for quantification.

### 2.5. Immunofluorescence

Paraffin cross-sections were subjected to deparaffinization and rehydration first. Antigen retrieval was performed in sodium citrate buffer (10 nM sodium citrate, 0.05% Tween-20, pH 6.0) at 90 °C for 20 min. The sections were then blocked and permeabilized in 5% donkey serum (NDS)/0.1% Triton X-100/1x PBS for one hour at room temperature (RT) and incubated with primary antibody for 18 h at 4 °C. The following day, the sections underwent secondary antibody incubation for one hour at RT. In addition to primary and secondary antibodies, we labeled the sections with Collagen Hybridizing Peptide, CHP (FLU60, 3Helix, Salt Lake City, UT, USA), following manufacturer instructions.

The cross-sections were then stained with 0.5% Sudan black to dampen down photoreceptor autofluorescence. Then, 4′,6-diamidino-2-phenylindole counterstaining was implemented to visualize the nuclei (DAPI; Thermo Fisher Scientific, Waltham, MA, USA). The slides were then mounted with ProLong Gold Antifade reagent (Thermo Fisher Scientific, Waltham, MA, USA). Nikon W1 Dual cam spinning disk confocal laser microscope system (Nikon, Tokyo, Japan) at the Center for Advanced Microscopy/Nikon Imaging Center of Northwestern University was used to image the section staining. The images were then analyzed, and Image J software (Image J, 1.54d, National Institutes of Health, Bethesda, MD, USA) was used for quantification of fluorescence intensity.

### 2.6. Retinal Flat-Mount Analysis

Eyes were fixed in 4% PFA/PBS at room temperature for 2 h. The retinas were then blocked with 5% normal donkey serum/1% bovine serum albumin (BSA)/0.3% Triton X-100/1x PBS for 1 h at room temperature. They were incubated with primary antibody for 18 h at 4 °C, followed by secondary antibody incubation for one hour. A total of 4 radial incisions were then performed to flatten the samples on a slide, and then we mounted the latter with ProLong Gold Antifade reagent (Thermo Fisher Scientific, Waltham, MA, USA). Nikon W1 Dual CAM spinning disc confocal laser microscope at the Northwestern University Center for Advanced Microscopy was used to image the samples. To calculate the vessel tortuosity, we used ImageJ software (Image J, 1.54d, National Institutes of Health, Bethesda, MD, USA). We calculated two measurements: the actual length of the vessels from a starting point or a branching point, which is calculated following the actual path of the vessels, and the Euclidean distance length, which is a second line from the starting point of the vessels straight to the end without following the path of the vessels. The ratio of actual length/Euclidean distance length gave us a measurement of the vessel tortuosity [9].

We also calculated the avascular area percentage in the peripheral retina using Image J (Image J 1.54d, National Institutes of Health, Bethesda, MD, USA).

### 2.7. Western Blot

Retinal tissues were lysed in RIPA buffer (R0278, Sigma-Aldrich, Munich, German) supplemented with protease and phosphatase inhibitors (11836170001 and 04906845001, Roche, Manheim, Germany) for 20 min on ice. The samples were then homogenized using the Tissue Lyser LT (Qiagen, Hilden, Germany) for 10 min at 4 °C. Lysates were cleared by centrifugation. Protein concentration was measured by Bradford assay (23238, Thermo Scientific, Waltham, MA, USA). A total of 30 mg proteins were denatured and separated on a 3–8% Tris-Acetate gel (EA0375BOX, Invitrogen, Waltham, MA, USA) for 1 h at 150 V, followed by transfer onto a nitrocellulose membrane for 1.5 h at 80 V at 4 °C. Membranes were blocked in 5% milk at room temperature (RT) for one hour, then incubated with primary antibodies for one hour at RT and then overnight at 4 °C. Membranes were washed in 0.1% TBS-T and then incubated with the HRP-conjugated antibodies for two hours at RT. See Table 1 for primary and secondary antibodies used. Immunoreactive bands were visualized with a chemiluminescent substrate (34577, Thermo Scientific, Waltham, MA, USA), and the iBright1500 imager (Invitrogen, Waltham, MA, USA) was used to acquire the images.

### 2.8. Optical Coherence Tomography (OCT) and Optical Coherence Tomography Angiography (OCTA)

Spectralis OCT2 system (Heidelberg Engineering, Heidelberg, Germany) was used to perform OCT and OCT-angiography imaging. A ketamine/xylazine solution was used to anesthetize the mice; then, they were given meloxicam for pain prophylaxis. After their eyes were dilated, we placed a contact lens (3.2 mm diameter, 1.7 mm base curve) on the surface of the right eye (90642, Cantor and Nissel Ltd., Brackley, UK). OCT2 system used infrared imaging to focus upon the optic nerve, then OCT (detail scan, 50 averaged frames) or OCT-angiography (high resolution, 7 averaged frames) was performed.

### 2.9. Statistics

GraphPad Prism version 9 software (GraphPad Software) was used to perform statistical analysis. A 2-tailed Mann–Whitney test was performed, and the data are reported as mean ± Standard Error of Measurement (SEM). Statistical significance was defined as *p* < 0.05.

## 3. Results

### 3.1. Retinal Morphology in LHIPR Model

We followed the schematic protocol shown in Figure 1A, described in the method section, to mimic LHIPR. We then stained the cross-sections with hematoxylin and eosin (H&E) and examined the retinal morphology at 1.5 months, 2 months, and 2.5 months. As shown in the representative pictures in Figure 1B, the retinal stratification appeared well preserved in LHIPR when compared to room air (RA) samples. We measured the thickness of each layer, and, as shown in the graphs in Figure 1B, we observed the inner plexiform layer (IPL) thinning at 1.5 months and 2 months but not at 2.5 months. This change is consistent with our previous data, where we observed a thinning of IPL at P12, P17, and P21 [8]. We also detected outer nuclear layer (ONL) thinning at all three time points analyzed in LHIPR retinas, showing outer retinal disruption, which we did not see in our previous comparisons up to P30.

### 3.2. Delayed Retinal Vascular Development in LHIPR Model

We analyzed the retinal vasculature by retinal flat mounts labeled with Isolectin B4 (IB4). In the previous paper, we observed an irregular vascularization at P12 in LHIPR, and by P30, the vasculature failed to completely develop [8]. We wanted to verify, at a later point, if the vasculature reached the retinal peripheral area. We observed a persistent delay in the vasculature formation in the peripheral area at 1.5 months when compared to the RA flat mount (Figure 2A). Even though we did see a progression in the vascularization process, we still observed a not complete vascular development at 2.5 months as well (Figure 2A). We quantified the avascular area percentage in the peripheral retina both at 1.5 months (18%) and 2.5 months (8%), as reported under the images in Figure 2A. Moreover, LHIPR vessels appeared tortuous and dilated at both analyzed time points (Figure 2A, white insert and graph).

### 3.3. Preretinal and Retinal Fibrosis Progression in LHIPR Model

We have previously shown substantial preretinal scar tissue formation in the P30 LHIPR retina, compared to RA and to previous time points [8], so here we wanted to further analyze this process at later time points. To study the fibrogenic process in the preretinal membrane, we labeled the cross-sections with the Collagen Hybridizing Peptide, CHP, a molecule that is interposed between newly forming collagen helixes [10,11] and conjugated with a green fluorophore. Therefore, the newly formed collagen is green in the figure. Collagen accumulation was seen at 1.5 months in LHIPR (Figure 2B) when compared to RA tissues, with a progressive increase in CHP labeling in the preretinal space at 2 months and at 2.5 months, when we started observing a fibrovascular membrane on top of the retina, as indicated by the arrowheads (Figure 2B). We then stained the cross-sections at 2.5 months for Collagen VI (COL6), another fibrosis marker. We found a positive signal in LHIPR retinas, while no staining was detected in control samples (Figure 2C). The COL6 signal was also seen in long vertical lines, suggesting expression in activated Müller cells. We confirmed this hypothesis by immunostaining for GFAP, which showed a co-localization with COL6 in the vertical lines (Figure 2C).

For an in vivo cross-sectional view of the retina, we performed optical coherence tomography (OCT). Starting from 1.5 months, we detected a membrane growing on the retinal surface, as shown in Figure 3A. At 2.5 months, the membrane was in the preretinal space, but we did not see similar lesions in the control retina. During later time points, and as shown in Figure 3A, membranes were evident in the preretinal space at 3, 4, and 5 months, and as indicated with the arrowheads in Figure 3A, they were associated with focal retinal traction. OCT Angiography (OCTA) performed at 4.5 months (Figure 3B) showed the presence of blood flow within these membranes in the preretinal space (yellow labeling in the figure), confirming their fibrovascular nature. Both the transverse and orthogonal images show big blood vessels perfusing the membrane growing on the surface of the retina.

### 3.4. Fibrosis and Abnormal Angiogenesis Progressively Increase at the Later Time Points

At the last observation time point of 9 months, we could visualize a vascularized membrane exerting traction on the central retina in LHIPR. At this time point, retinal detachment was also visible in the LHIPR tissue when compared to the room air samples in cross-sections stained with H&E. More details are shown in the higher magnification in Figure 4A (arrowheads). In addition, OCT Angiography (OCTA) confirmed the presence of blood flow within these membranes in the preretinal space (yellow labeling, Figure 4B), confirming the existence of perfused vasculature in this membrane.

We then stained the cross-sections with CHP and, as shown in Figure 5A, staining consistent with newly formed collagen was observed in LHIPR retinal tissue when compared to controls, where, as expected, the strongly positive sclera was the only structure labeled (Figure 5A(I)). In LHIPR, the newly formed collagen was observed in the posterior retina, likely in hyaloid remnants, consistent with the histological analysis, and confirming the fibrotic nature of this preretinal tissue as indicated by the arrowheads (Figure 5A(II–IV)). This newly formed collagen was also observed around the lens.

Next, we stained the cross-sections for COL6 and observed significant staining on the surface of the retina in LHIPR, as well as intraretinal staining in long vertical streaks, colocalizing with Müller cells (Figure 5B(I,II)). At this time point, retinal staining involved the inner and outer retina, compared to 2.5 months, where it was largely outer; a higher magnification of the long vertical streaks is shown (Figure 5B(III)). Moreover, double immunostaining for GFAP and COL6 confirmed the expression of the fibrotic marker in the activated Müller cells (Figure 5B(IV–VI)), and as shown with the arrowhead in B-V, we detected GFAP+ astrocyte showing COL6 expression.

To quantify fibrosis, we then performed retinal Western blot analysis for COL6 (Figure 5C). No band was observed in the RA (control) samples, whereas a very strong and specific band was present in all three LHIPR samples analyzed, confirming the upregulation of the protein in the disease model. We next evaluated collagen 1 (COL1) and detected expression in the LHIPR retinal samples (Figure 5C). For COL1, the antibody detected three bands: the pro-collagen band (highest molecular weight, 200 kDa), the mature collagen band (130 kDa), and the c-terminal pro-peptide band (lower molecular weight, 37 kDa).

### 3.5. Retinal Inflammation in LHIPR Model

Next, we wanted to expand our understanding by exploring other pathological long-term consequences beyond fibrosis and angiogenesis in this model. To study the inflammatory processes at 9 months, we labeled the cross-sections for the monocyte-derived macrophage marker LY6C in combination with the endothelial cell marker CD31. Compared to controls, where little to no signal was observed, LY6C+ macrophages were abundant in LHIPR tissue (Figure 6A). In addition, adjacent LY6C+ and CD31+ cells were found in the preretinal space along the fibrovascular membrane (arrowhead). This localization is more evident at higher magnification, as shown in the insert for Figure 6A.

In addition, we labeled the 9-month cross-sections for resident macrophage markers F480 and IBA-1. We detected a strong signal in the LHIPR samples and no signal in the control samples. A higher magnification of the positive cells in LHIPR sections is provided in Figure 5B, where the arrowhead indicates the fibrovascular membrane. These results confirm a prominent inflammatory response at 9 months in the LHIPR retina.

## 4. Discussion

This study expands the long-term findings in the LHIPR model, focusing on the fibrovascular membranes and inflammation. This study allowed us to further characterize this rodent model in the context of ischemic retinopathies, showing persistent inflammation, fibrosis, and abnormal vascular proliferation up to 9 months of age.

We had previously demonstrated that limited high oxygen exposure from P0 to P7 in LHIPR reproduces several characteristics of human ROP [8,12,13]. Moreover, we showed the prolonged course of retinopathy without proof of vascular repair and revascularization, unlike the OIR model, making it a useful model for understanding the pathological progression and potential therapeutic interventions addressing fibrovascular consequences of ischemic retinopathies [8]. In this report, we further explore the long-term course of this model for 9 months, with intermediary time points, mainly focusing on fibrovascular membrane formation. Overall, the retinal morphology was relatively preserved at the earliest time points (1.5 months, 2 months, and 2.5 months), although later, we observed traction by the fibrovascular preretinal membrane and retinal detachment in the LHIPR tissue at 9 months compared to the control samples. This was the likely explanation for outer nuclear layer thinning, which we observed as early as 1.5 months, suggesting outer retinal pathological insult related to chronic retinal separation.

Retinal detachment occurs in the advanced stages of ROP, and it is mostly due to the extraretinal fibrovascular proliferation associated with stages 4 and 5 ROP, which represent the advanced fibrovascular phase of ROP [2]. Fibrovascular membrane proliferation is also a pathological manifestation of other ischemic retinopathies and one of the main drivers of the late stage of advanced proliferative diabetic retinopathy (PDR), responsible for the development of diabetic traction retinal detachments and vision loss [3]. Therefore, a better understanding of this process is strongly needed. Using OCT, we followed the progression of the fibrovascular membrane over time in the LHIPR model, characterizing progression between 1.5 and 9 months of age. The fibrotic nature of the preretinal tissue was confirmed by staining the cross-sections with CHP and COL6, where a strong signal was detected over time. CHP is a peptide that specifically binds to the degraded, unfolded collagen chains. Relevant to our study, CHP has been used for quantifying fibrosis in human samples and in mouse models. In a pilot study of human samples of liver fibrosis, this method was used to evaluate the fibrosis extent [10]. In another study, CHP was shown to bind the newly formed collagen, demonstrating extracellular matrix remodeling at very early stages of liver fibrosis in human specimens and in a mouse model of induced chronic liver fibrosis [10]. Moreover, a recent study has used CHP for quantifying fibrosis in choroidal neovascularization (CNV). Using two different mouse models, the laser-induced CNV and the JR5558 spontaneous CNV, CHP detected subretinal fibrotic areas in vivo and ex vivo. In that study, the authors demonstrated co-expression of CHPs with fibrosis markers, such as fibronectin, with significantly decreased expression after using a bispecific angiopoietin-2/vascular endothelial growth factor-A antibody [14].

COL6 is an interstitial collagen, a structural component of the extracellular matrix that has also been described as a profibrotic molecule, representing a potential target for multiple fibrotic disorders. Studies suggest that COL6 might be a major regulator of myofibroblast lineage, modulating myofibroblast contractility, extracellular matrix deposition, chemotaxis, and wound healing. COL6 expression was seen in fibrotic conditions, such as idiopathic pulmonary fibrosis (IPF), where it was distinguishable from COL1 and correlated with α-SMA positive myofibroblasts in the fibrotic foci in human specimens. Its expression was also elevated in chronic liver diseases such as hepatic fibrosis and hepatocellular carcinoma, as well as in diabetic glomeruli, both in humans and mice, as shown by immunohistochemical localization studies [15,16,17]. Moreover, fragments from the C-terminal of the α3 chain of collagen 6 have been described as having macrophage chemoattractant properties, suggesting that collagen 6 may potentially be involved in inflammatory signaling in addition to its structural functions [18,19,20,21].

In the eye, collagen 6 was shown to be expressed in physiological as well as pathological conditions. Its expression, in particular COL6A1, was detected in the internal limiting membrane (ILM) in freshly isolated human retinectomy samples [22]. It is also physiologically expressed in the human cornea, and its expression levels do not change in keratoconus corneal sections, except for scar tissue areas in some keratoconus samples, suggesting its involvement in corneal fibrogenesis [23]. Immunostaining in preretinal membranes isolated from diabetic retinopathy patients showed COL6A3 accumulation in pathological tissue [24]. In addition, RNASeq analysis of human CNV membranes showed a significant upregulation of *Col6a1* when compared to age-matched macular retinal pigment epithelium and choroidal tissue [25].

Interestingly, in addition to its expression in the fibrovascular membrane, we detected COL6 signal in the Müller cell compartment of the LHIPR retina as early as 2.5 months, with a stronger signal at 9 months, suggesting potential activation and involvement of these glial cells in fibrogenesis. Müller glial–mesenchymal transition has been described, and Müller cells undergo reactive gliosis, triggering cell proliferation and protruding cytoplasmic extensions, both contributing to epiretinal scar formation [26]. Therefore, we speculate that this mechanism could be involved in the fibrovascular process of LHIPR.

Pathological angiogenesis characterizes the vascular phase (phase 3) of ROP and is also characteristic of advanced PDR. Unlike physiological angiogenesis, which is finely modulated by precise gradients of angiogenic factors and molecules, the unbalanced overproduction of angiogenic factors during late ROP and proliferative retinopathies triggers pathological retinal processes leading to disorganized preretinal neovascularization. We performed OCTA to confirm the presence of blood vessels in the preretinal membrane of the LHIPR model, which showed blood flow within the membrane at 4.5 and 9 months, the latest time point in our study, supporting the fibrovascular nature of this tissue. Moreover, retinal flat mounts at 1.5 and 2.5 months showed large retinal vessels, likely extensions from the original hyaloidal vasculature. As described in our prior study, the hyaloidal vasculature was no longer visible in the room air group, whereas it remained prominent until P30 in the LHIPR model. We also observed that hyaloidal vessels invaded the central and peripheral retina at P12 and P17, which may reflect the origin of these persistent, abnormally located vessels in the fibrovascular membranes [8].

Our results also show delayed retinal vascular development in the LHIPR model and the persistence of peripheral avascular area in the retina. This is particularly relevant in our model since a recently published study investigated the vascular abnormalities in patients with regressed or treated ROP through a follow-up until school age, showing that one third of the ROP eyes had persistent avascular retina [27].

Angiogenesis is deeply modulated by a complex network of cytokines, extracellular matrix components, and growth factors. Inflammatory cytokines can modulate angiogenesis and fibrosis as well. A strong inflammatory activity was seen in the fibrovascular membrane of LHIPR, as shown by the immunofluorescence for F4/80, IBA-1, both resident macrophage markers, as well as LY6C, which reflects monocyte-derived macrophage infiltration in the tissue. The presence of these diverse cell types suggests there is an active inflammatory process in the preretinal membrane. Fibrosis, angiogenesis, and inflammation, as seen in this model, are also important features in fibrovascular membranes surgically removed from PDR patients, highlighting important similarities of this model to human ischemic retinopathies. Transcriptomic analysis at a single cell level in the human fibrovascular tissue, followed by in vitro validation experiments, identified immune cells, stromal cells, and endothelial cells as the main cell populations, with stromal cells exhibiting prominent fibrogenic properties [28]. In addition, endothelial cells in these PDR membranes were actively involved in angiogenesis and showed transcriptomic irregularities. A cluster of pro-angiogenic macrophages among the immune cell compartment was found, confirming the role of inflammation in these fibrovascular membranes and the active involvement of immune cells in the formation of the fibrovascular lesions. The current work shows that macrophage inflammatory cells are prominent in the fibrovascular tissue in LHIPR, supporting the relevance of this model to human proliferative retinopathy.

## 5. Conclusions

In conclusion, our data confirm the LHIPR model presents a fibrovascular component that could be harnessed to understand the pathogenesis of these traction membranes in ROP and other ischemic retinopathies, including proliferative diabetic retinopathy. Further investigation into the molecular mechanisms underlying these processes will be critical for identifying new therapeutic approaches to prevent and ameliorate the devastating visual consequences of these membranes in patients with advanced complications of ischemic retinopathies.

## Figures and Tables

**Figure 1 cells-12-02468-f001:**
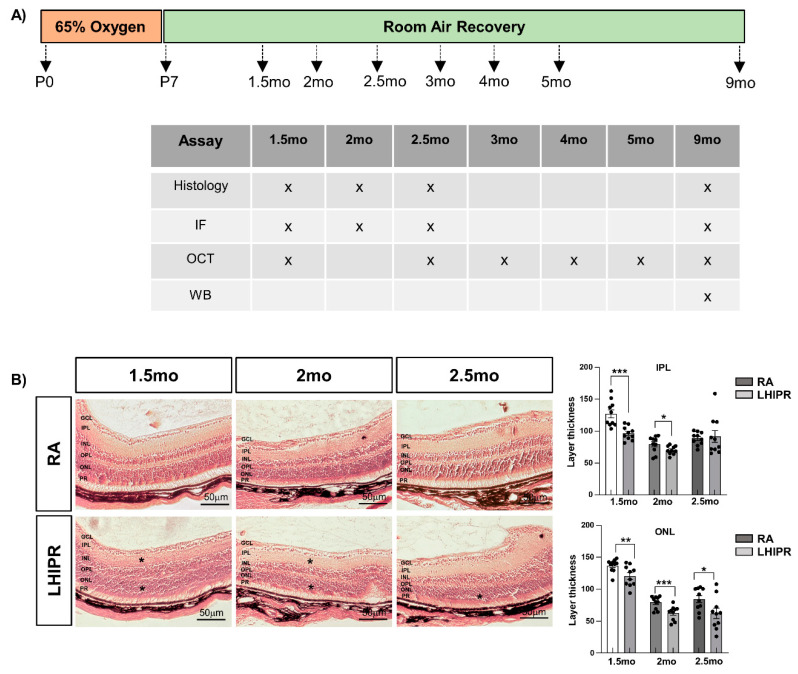
Limited hyperoxia-induced proliferative retinopathy (LHIPR)-induced structural and vascular changes at 1.5 months, 2 months, and 2.5 months. Schematic of LHIPR treatment of 65% oxygen from P0 to P7 followed by room air recovery until experimental endpoints is shown in (**A**). A table showing the assays performed at each time point is reported in A as well. H&E staining on paraffin-embedded histological cross-sections of 1.5-month, 2-month, and 2.5-month murine eyes in room air (RA) and LHIPR condition, and quantification of IPL and OPL thickness reported in the graphs are shown in (**B**). The retinal layers are labeled. Scale bar = 50 μm. Mann–Whitney test: * *p* < 0.05, ** *p* < 0.01, and *** *p* < 0.001.

**Figure 2 cells-12-02468-f002:**
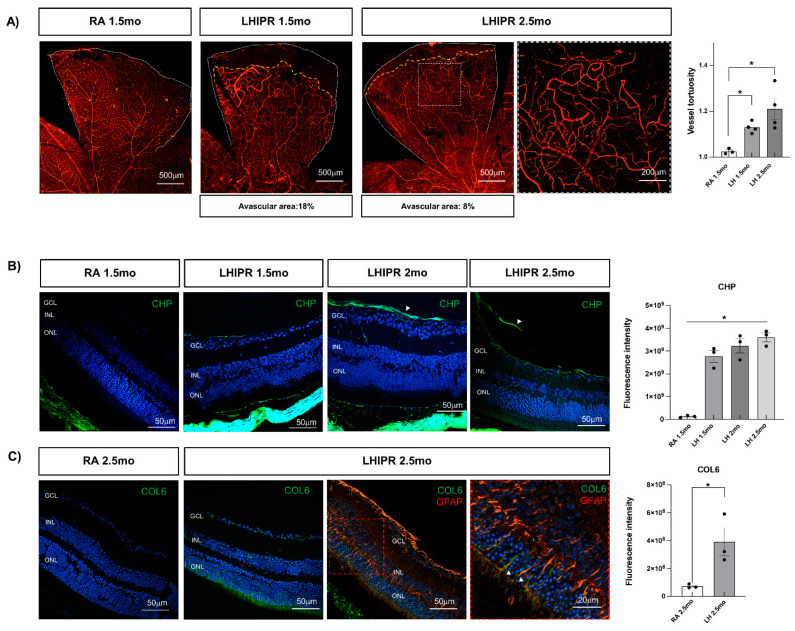
Representative images of retina flat-mount samples from both room air and LHIPR groups at 1.5 months and 2.5 months stained with IB4 antibody to detect vascular development (**A**). The white dotted lines delimit the retinal flat surface, whereas the yellow dotted lines delimit the beginning of the avascular area that was quantified. The avascular area percentage is reported under the images in (**A**). Higher magnification shows tortuous vessels at 2.5 months in LHIPR (**A**). The graph on the right shows the vessel tortuosity calculated as actual length/Euclidean distance length. CHP staining of paraffin-embedded histological cross-sections at 1.5 months, 2 months, and 2.5 months in LHIPR condition compared to RA staining as shown in (**B**). The arrowheads indicate the accumulation of CHP and the fibrovascular membrane at 2.5 mo. A graph showing the CHP intensity of fluorescence is reported on the right. COL6 labeling on paraffin-embedded histological cross-sections of 2.5-month murine eyes in room air (RA) and LHIPR is shown in C. A graph showing the CHP intensity of fluorescence is reported on the right. Co-staining of COL6 with GFAP in LHIPR group is shown in (**C**) as well; arrowheads in higher magnification images indicate the COL6+/GFAP+ cells. DAPI was used to counterstain the nuclei. The retinal layers are labeled. Scale bar = 20 μm, 50 μm, 200 μm, and 500 μm. Representative images from 3 independent experiments are shown. Mann–Whitney test: * *p* < 0.05.

**Figure 3 cells-12-02468-f003:**
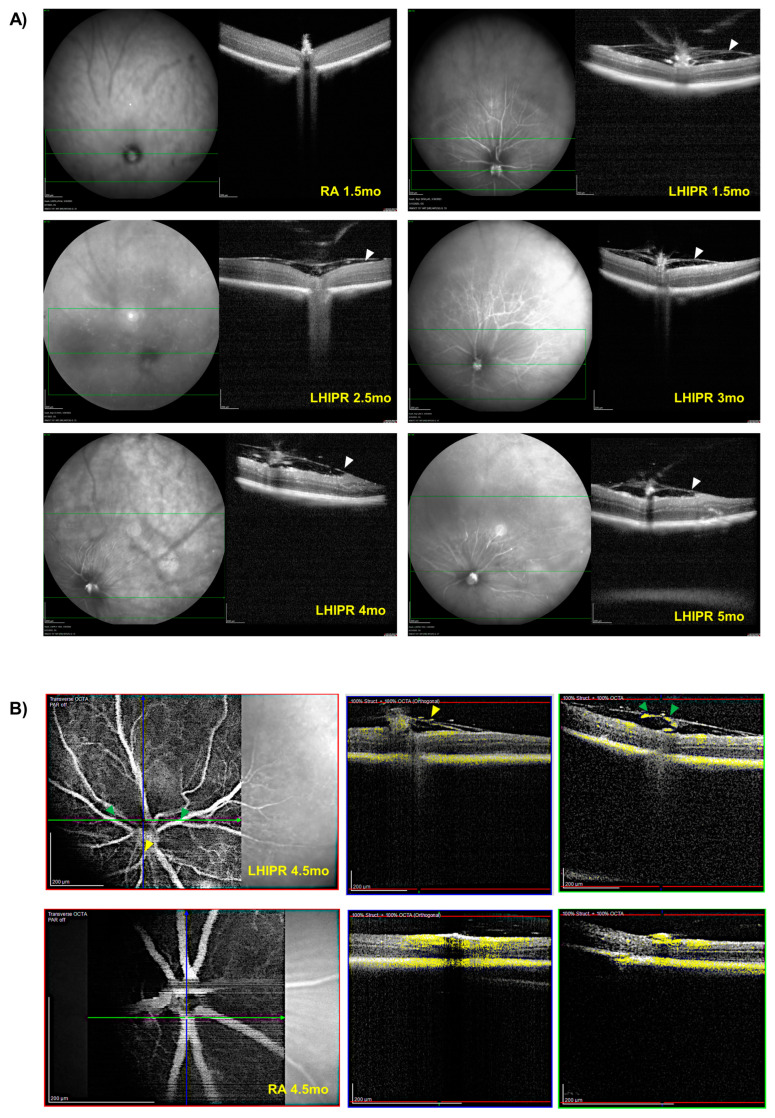
OCT analysis on LHIPR mice during the progression of the disease. Representative images from OCT scan at RA 1.5 months and LHIPR 1.5 months, 2.5 months, 3 months, 4 months, and 5 months showing the formation of the fibrovascular membrane over time are shown in panel (**A**). Arrowheads indicate focal retinal traction. OCTA scan of RA and LHIPR at 4.5 months are shown in (**B**). Both the transverse and orthogonal images are shown (blue and green boxes). The yellow flow signal highlights the presence of blood flow in the fibrovascular membrane on cross-section. The yellow arrowheads indicate the vessels in the orthogonal image, and the green arrowheads indicate the vessels in the transverse image. Scale bar = 200 μm.

**Figure 4 cells-12-02468-f004:**
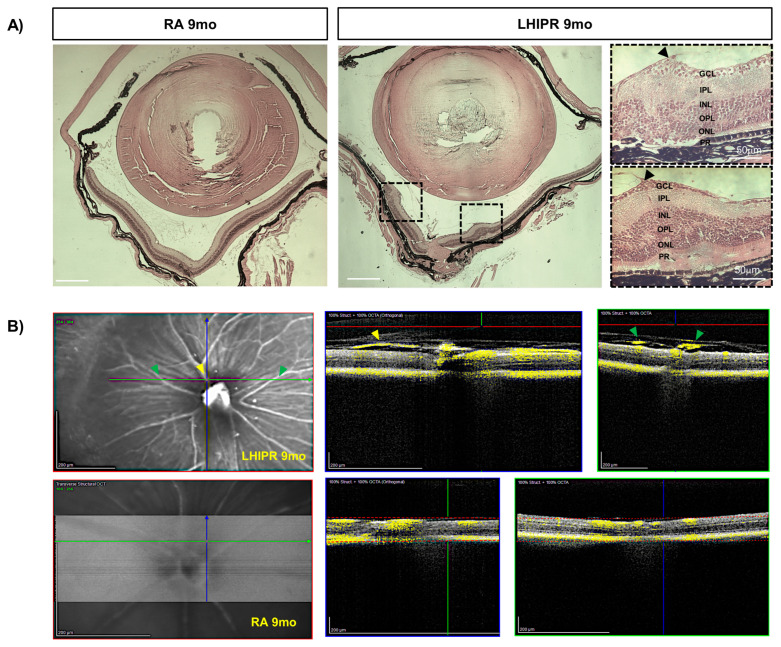
Retinal pathological changes in LHIPR at 9 months. H&E staining on paraffin-embedded histological cross-sections of 9-month murine eyes in room air (RA) and LHIPR condition (**A**). Retinal detachment is also visible. The arrowheads in the higher magnification squares on the right of panel (**A**) indicate the pulling preretinal membrane on both sides of the hyaloid reminiscence. The retinal layers are labeled. Scale bar = 50 μm (**A**). OCTA representative images at 9 months in RA and LHIPR (**B**). Both the transverse and orthogonal images are reported. The yellow flow signal highlights the blood flow in the fibrovascular membrane on cross-section (arrowhead). The yellow arrowheads indicate the vessels in the orthogonal image, and the green arrowheads indicate the vessels in the transverse image (**B**). Scale bar = 200 μm.

**Figure 5 cells-12-02468-f005:**
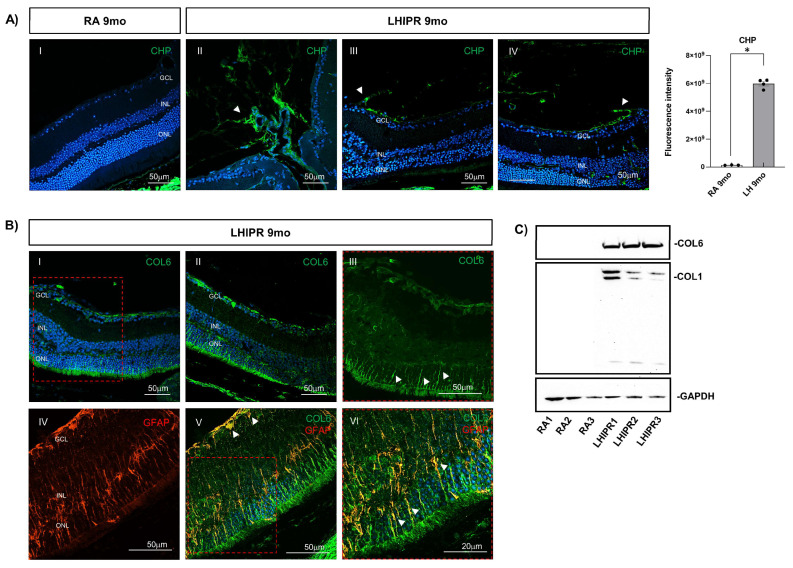
Fibrosis exacerbation in LHIPR at 9 months. Representative immunofluorescence images of CHP staining on paraffin-embedded histological cross-sections of 9-month murine eyes in LHIPR condition (II, III, and IV) compared to RA (I) are shown in (**A**). The arrowheads indicate the CHP+ fibrovascular membrane. In LHIPR retinas, we observed the presence of a fibrovascular membrane on the surface of the retina. The newly formed collagen was observed in the posterior retina, likely in hyaloid remnants, consistent with the histological analysis and confirming the fibrotic nature of this preretinal tissue. A graph showing the CHP intensity of fluorescence is reported on the right. Mann–Whitney test: * *p* < 0.05. COL6 staining on paraffin-embedded histological cross-sections of 9-month murine eyes in LHIPR condition is shown in (**B**) (I–III). Higher magnification of COL6 staining is shown in (**B**) (III), where arrowheads indicate the COL6+ cells. GFAP labeling and co-staining with COL6 on paraffin-embedded histological cross-sections of 9-month murine eyes in the LHIPR group is shown in (**B**) (IV–VI). An insert showing a higher magnification of an area of interest is reported in VI. Arrowheads indicate the COL6+/GFAP+ cells. DAPI was used to counterstain the nuclei. The retinal layers are labeled. Scale bar = 20 μm, 50 μm. The Western blot analysis for COL6 and COL1 at 9 months in RA samples compared to LHIPR samples is reported in (**C**). RA N = 3 and LHIPR N = 3. GAPDH was used as a loading control.

**Figure 6 cells-12-02468-f006:**
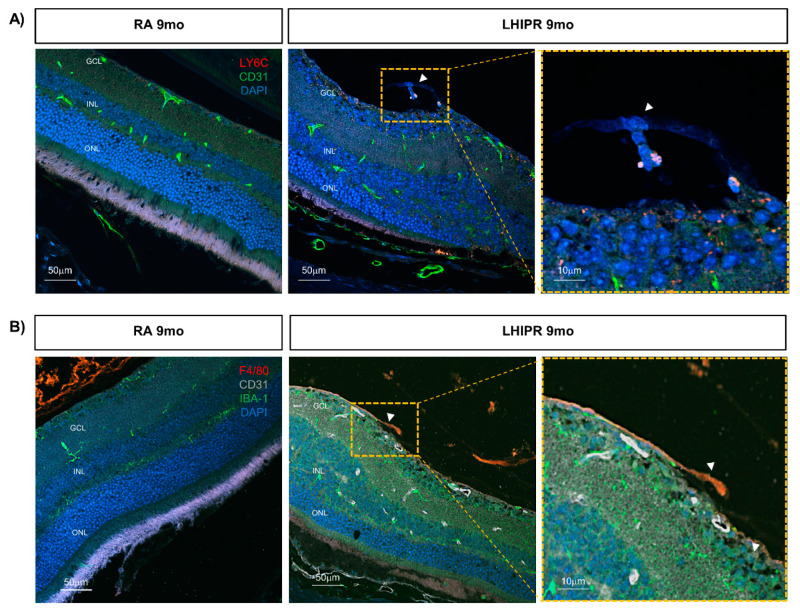
Inflammation in LHIPR at 9 months. Representative immunofluorescence images of the monocyte-derived macrophage marker LY6C and the endothelial marker CD31 labeling on paraffin-embedded histological cross-sections of 9-month murine eyes in LHIPR condition compared to RA are shown in (**A**). On the right, the insert with higher magnification of an area of interest is shown (the arrowhead indicates the fibrovascular membrane). Representative immunofluorescence images of the macrophage markers F480 and IBA-1 with the endothelial marker CD31 labeling on paraffin-embedded histological cross-sections of 9-month murine eyes in LHIPR condition compared to RA are shown in (**B**). On the right, the insert with higher magnification of an area of interest is shown (the arrowhead indicates the fibrovascular membrane). DAPI was used to counterstain the nuclei. The retinal layers are labeled. Scale bar = 10 μm, 50 μm.

**Table 1 cells-12-02468-t001:** List of antibodies used for immunofluorescence and Western blot.

Target	Source	Company	Catalog #	Dilution	Detection
COL6	Rabbit	Abcam (Cambridge, UK)	ab182744	1:50	IF
LY6C	Rat (avidin)	Abcam (Cambridge, UK)	ab15674	1:500	IF
F480	Rat	Abcam (Cambridge, UK)	ab16911	1:100	IF
CD31	Goat	R&D systems (Minneapolis, MN, USA)	AF3628	1:500	IF
IBA-1	Rabbit	Wako (Richmond, VA, CA)	019-19741	1:500	IF
GFAP	Rabbit	Invitrogen (Waltham, MA, USA)	PA5-90894	1:500	IF
COL6	Rabbit	Abcam (Cambridge, UK)	ab182744	1:1000	WB
COL1	Rabbit	Abcam (Cambridge, UK)	ab260043	1:1000	WB
GAPDH	Mouse	Invitrogen (Waltham, MA, USA)	MA5-15738	1:5000	WB
Isolectin GS-IB4-Alexa568	-	Thermo Fisher Scientific (Waltham, MA, USA)	I21412	1:50	FM
Secondary Ab	Source	Company	Catalog #	Dilution	Conjugate
Rabbit	Donkey	Jackson ImmunoResearch (West Grove, PA, USA)	711-545-152	1:200	AlexaFluor 488
Goat	Donkey	Jackson ImmunoResearch (West Grove, PA, USA)	705-605-147	1:200	AlexaFluor 647
Streptavidin	-	Jackson ImmunoResearch (West Grove, PA, USA)	S32356	1:200	AlexaFluor 594
Rat	Donkey	Jackson ImmunoResearch (West Grove, PA, USA)	712-295-153	1:200	AlexaFluor 568
Rabbit	Goat	Thermo Fisher Scientific (Waltham, MA, USA)	32460	1:10,000	HRP
Mouse	Goat	Thermo Fisher Scientific (Waltham, MA, USA)	31430	1:10,000	HRP

## Data Availability

Data sharing is not applicable to this article.

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
