# Peer review of "Limited Hyperoxia-Induced Proliferative Retinopathy (LHIPR) as a Model of Retinal Fibrosis, Angiogenesis, and Inflammation"

_cells, 2023, doi:10.3390/cells12202468_

Round 1

Reviewer 1 Report

This manuscript reports on studies to extend the authors’ previous analysis of retinal pathology in the Limited Hyperoxia Induced Proliferative Retinopathy (LHIPR) model by characterizing the fibrovascular response over an extended time period. Development of this model is important in that it demonstrates severe pathologies seen in retinopathy of prematurity and advanced stages of diabetic retinopathy, such as preretinal fibrotic scar formation, tortuous retinal vessels, and persistent hyaloidal vessels.  Because these pathologies do not occur in other models of ischemic retinopathy, the LHIPR model can be used to develop a better understanding of the pathogenesis of retinal fibrosis and traction detachment. Using OCT, the authors characterized progression in the LHIPR model between 1.5 and nine months of age. The fibrotic nature of the preretinal tissue was confirmed by staining with CHP and COL6, where a strong signal was detected that increased over time. The studies also showed Collagen 6 expression in the fibrovascular membrane as well as in Müller cells of the LHIPR retina, as early as 2.5 months with a stronger signal at 9 months, suggesting potential activation and involvement of the Muller cells in fibrogenesis. Further studies in this model could be useful for identifying new therapeutic approaches to prevent and ameliorate the development of this pathology in patients with advanced complications of ischemic retinopathies.

While the observations are important and the results are generally well described, there are areas where improvement is needed.

The text describes a persistent delay in retinal vascular development. However. this is not obvious in the images shown in Figure 1C. A quantitative analysis should be performed to demonstrate this convincingly.

The suggestion of increased vascular tortuosity and dilation should be verified by quantitative analysis.

Quantitative analysis should also be performed to verify the suggested increases in immunoreactivity for Collagen hybridizing peptide and Collagen VI.

Please label the images in Figure 2A to clearly identify the membrane growing on the retinal surface.

The images presented in Figure 2B that are said to show blood flow within the pre-retinal membranes are not convincing and should be improved.

Please label the retinal images to show the different retinal layers and provide arrows and arrowheads to indicate the specific areas referred to in the text and figure legends.

The western blot shows striking differences in levels of mature Collagen 1 in the 3 independent samples. What is the meaning of this result?

Additional evidence should be provided to support the suggested increase in retinal inflammation in the LHIPR model. Increases in IBA-1 and F4/80 immunoreactive profiles suggest that numbers of macrophage and/or microglial cells are increased in the LHIPR retinas, but neither marker is specific for inflammation.

The legend for Figure 4 is confusing. Please revise the text, label the layers, and provide arrows and/or arrowheads to point out all features being noted. Please outline the area where the inset is taken from.

Please label layers in Figure 5 and add arrows to indicate the “growing fibrovascular membrane”. Also, what is the evidence that this is a growing membrane? A higher magnification/high resolution image would be more convincing.

Please review the text carefully to check for missing words and letters.

Please edit the abstract to explain the importance of fibrosis and traction for readers who may not be aware of these clinical aspects of retinopathy and to provide more experimental detail (i.e., what parameters were measured? What time points were checked, etc.)

The quality of the English language is good but there are some missing words and letters throughout the text. The authors should use a spelling and grammar checker to correct the text.

Reviewer 2 Report

This study describes a Limited Hyperoxia-Induced Retinopathy mouse model that shows the pathological phenotypes that occur in Retionopathy of Prematurity (ROP) and other ischemic retinopathies. In the field of ophthalmology, these retinopathies pose significant challenges in detection and management of these disorders. Data presented on the LHIPR model demonstrate that this model has potential for use to evaluate the molecular mechanism of pathogenesis in ischemic retinopathies which will be critical for developing new therapeutic approaches to prevent the advanced complications of ischemic retinopathies.

Revisions needed:

Line 123 – …4 C – change to 4°C

Line 128 – …4 C – change to 4°C

Line 131 – were visualized (insert with) a chemoluminscent…

Line 148 – include the words Standard Error of Measurement (SEM).

Line 154 – delete the word the before Figure 1B.

Line 164 – Figure 1 – the images in Figure 1 B, and C-E are very small. Consider breaking up this figure with C, D and E covering a whole page.

Line 166 – change shown to showing, insert at after performed to read performed at each time point…

Line 183 – after verify insert at a to read at a later point.

Line 217 – delete the word the before panel A.

Line 233 – delete the word the before panel (A); Change the word side to read sides…

Line 246 – change the word months to month...

Line 248 – change the word months to month...

Line 250 – change the word months to month...

Line 255 – insert the word a after as to read as a loading control.

Line 266 – remove the word the before three LHIPR samples…

Line 270 – remove the space after the ( and before 130 kDA)..

Line 280 – Figure 5 images are also very small. Consider enlarging these images.

Line 283 – change the word months to month...

Line 290 – change the word months to month...

Line 317 – Write out PDR – Proliferative Diabetic Retinopathy.

Line 334 – Westenskow, et al should be listed in the references.

Line 390 – change the word that to there is…should read suggests there is an active….

Line 394 – insert a after at and before single…should read at a single cell level…

Line 397 – Corano Scheri paper should be listed in the references.

Line 399 – change pro-angiogenetic to pro-angiogenic….

Minor corrections.

Round 2

Reviewer 1 Report

The authors have addressed some of the concerns raised in my previous review. However, several important issues require further attention.

1.  The text describes a persistent delay in retinal vascular development. Lines 182-188 states:

We analyzed the retinal vasculature by retinal flat mounts labeled with Isolectin B4. In the previous paper we observed an irregular vascularization at P12 in LHIPR and by P30 the vasculature failed to completely develop [8]. We wanted to verify, at a later point, if the vasculature reached the retinal peripheral area. We observed a persistent delay in the vasculature formation, in the peripheral area, at 1.5 months when compared to the RA flat mount (Figure 2A). Even thought we did see a progression in the vascularization process, we still observed a not complete vascular development at 2.5 months as well.

Because this delay in vascularization of the peripheral area is not clear in the images provided, I suggested that a quantitative analysis of that delay should be performed. However, the authors did not quantify the peripheral avascular area. Instead they measured retinal vascular density. That is not acceptable to prove a delay in vascularization. Better images of the peripheral avascular area should be presented and that area should be quantified. Relative decreases in vascular density do not prove a delay in vascularization.

6. Please label the retinal images to show the different retinal layers and provide arrows and arrowheads to indicate the specific areas referred to in the text and figure legends.

R. We labeled the histological retinal images to indicate all the retinal layers.

Retinal layers in all figures should be labeled.

R. We also provided asterisks to indicate the specific areas we discuss in the results.

No asterisks are evident in the revised figures.

9. The legend for Figure 4 (now 5) is confusing. Please revise the text, label the layers, and provide arrows and/or arrowheads to point out all features being noted. Please outline the area where the inset is taken from.

In Figure 5A, LHIPR 9mo panels are still unclear. What are the green fluorescent structures on the vitreous side of the image? The text should be expanded and arrows or asterisks should be added to clearly identify the pathological features.

In Figure 6, panels A and B, the arrowheads to indicate the membrane are missing.

Round 3

Reviewer 1 Report

The authors have revised as recommended.